# RETHINKING CLASS-PRIOR ESTIMATION FOR POSITIVE-UNLABELED LEARNING

**Yu Yao**[1]   **Tongliang Liu**[1][†]   **Bo Han**[2]   **Mingming Gong**[3]
**Gang Niu**[4]   **Masashi Sugiyama**[4,5]   **Dacheng Tao**[6,1]

[1]The University of Sydney   [2]Hong Kong Baptist University   [3]The University of Melbourne
[4]RIKEN AIP   [5]The University of Tokyo   [6]JD Explore Academy, China

## ABSTRACT

Given only *positive* (P) and *unlabeled* (U) data, PU learning can train a binary classifier without any *negative* data. It has two building blocks: PU *class-prior estimation* (CPE) and PU classification; the latter has been well studied while the former has received less attention. Hitherto, the distributional-assumption-free CPE methods rely on a critical assumption that *the support of the positive data distribution cannot be contained in the support of the negative data distribution*. If this is violated, those CPE methods will systematically *overestimate* the class prior; it is even worse that we *cannot verify* the assumption based on the data. In this paper, we rethink CPE for PU learning—can we remove the assumption to make CPE always valid? We show an affirmative answer by proposing Regrouping CPE (ReCPE) that builds an *auxiliary* probability distribution such that the support of the positive data distribution is never contained in the support of the negative data distribution. ReCPE can work with any CPE method by treating it as the base method. Theoretically, ReCPE *does not affect* its base if the assumption already holds for the original probability distribution; otherwise, it *reduces the positive bias* of its base. Empirically, ReCPE improves all state-of-the-art CPE methods on various datasets, implying that the assumption has indeed been violated here.

## 1 INTRODUCTION

*Positive-unlabeled* (PU) learning can date back to 1990s (Denis, 1998; De Comité et al., 1999; Letouzey et al., 2000), and there has been a surge of interest in this learning scenario in recent years because of the difficulty to annotate large-scale datasets (Ren et al., 2014; du Plessis et al., 2014; 2015; Christoffel et al., 2016; Jain et al., 2016; Ramaswamy et al., 2016; Sakai et al., 2018; Kato et al., 2018; Bekker & Davis, 2018; Gong et al., 2019; Bai et al., 2021; Xia et al., 2021; Yao et al., 2021). It is also fallen into different applications, such as knowledge-base completion (Galárraga et al., 2015; Neelakantan et al., 2015), text classification (Lee & Liu, 2003; Li & Liu, 2003), and medical diagnosis (Claesen et al., 2015; Zuluaga et al., 2011).

PU learning can be divided into two different settings based on different data generation processes. The first setting is called *censoring* PU learning (Elkan & Noto, 2008), which follows a one-sample configuration. Specifically, a sample $S$ is randomly drawn from the unlabeled data distribution $P_{\mathrm{u}}$, and a positive sample $S_{\mathrm{p}}$ is then distilled from it, i.e., randomly selecting some positive instances contained in the unlabeled data to be the positive sample. The second setting is called *case-control* PU learning (Kiryo et al., 2017). In this setting, a positive sample $S_{\mathrm{p}} = \{x_i\}_{i=1}^{k}$ is randomly drawn from the positive class-conditional distribution $P_{\mathrm{p}} = P(X|Y=1)$, and an unlabeled sample $S_{\mathrm{u}} = \{x_i\}_{i=k+1}^{n}$ is randomly drawn from the unlabeled data distribution $P_{\mathrm{u}}$. Because case-control PU learning is more general than censoring PU learning (Niu et al., 2016), therefore, we will focus on the setting of case-control PU learning.

Under the setting of case-control PU learning, a lot of classification methods have been proposed (Ren et al., 2014; du Plessis et al., 2014; 2015; Christoffel et al., 2016; Sakai et al., 2018; Kato et al., 2018; Bekker & Davis, 2018; Kwon et al., 2019; Tanielian & Vasile, 2019; Gong et al., 2019).

---

[†]Correspondence to Tongliang Liu (tongliang.liu@sydney.edu.au).

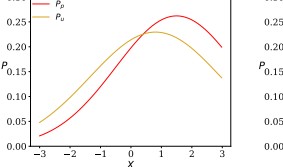 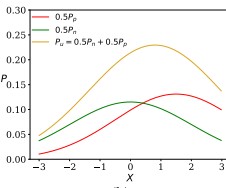 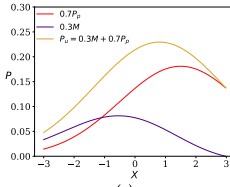 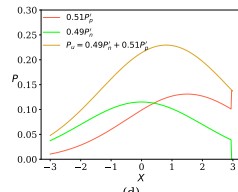

Figure 1: (a) The unlabeled data distribution $P_{\mathrm{u}}$ and the positive class-conditional distribution $P_{\mathrm{p}}$ are given. (b) Assume that the latent negative class-conditional distribution $P_{\mathrm{n}}$ is fixed, i.e., $0.5P_{\mathrm{n}}$ is shown by the green curve, and that the class-prior $\pi$ is 0.5, i.e., $P_{\mathrm{u}} = 0.5P_{\mathrm{n}} + 0.5P_{\mathrm{p}}$. (c) The existing distributional-assumption-free CPE methods will output $0.7$ instead of $0.5$ because they always output the maximum proportion $\kappa^*$ of $P_{\mathrm{u}}$ in $P_{\mathrm{p}}$. (d) Applying the proposed ReCPE method, a auxiliary distribution $P_{\mathrm{p}'}$ will be created and the existing CPE methods will output $\pi' = 0.49$ instead of $0.7$ with input $P_{\mathrm{p}'}$ and $P_{\mathrm{u}}$ instead of $P_{\mathrm{p}}$ and $P_{\mathrm{u}}$.

However, the *class-prior estimation* (CPE) (Elkan & Noto, 2008; Jain et al., 2016; Ramaswamy et al., 2016; Christoffel et al., 2016; Kato et al., 2018) has received less attention. Formally, CPE is defined as a problem of estimating $\pi = P(y = 1) \in (0, 1)$ given a sample from the marginal distribution $P_{\mathrm{u}}$ and a sample from positive class-conditional distribution $P_{\mathrm{p}}$. The marginal distribution $P_{\mathrm{u}}$ is mixed with both positive and negative class-conditional distributions, i.e., $P_{\mathrm{u}} = \pi P_{\mathrm{p}} + (1-\pi)P_{\mathrm{n}}$. CPE acts as a crucial building block for state-of-the-art PU classification methods, and it is essential to build statistically-consistent PU classifiers (du Plessis et al., 2014; Scott, 2015; Jain et al., 2016; Kiryo et al., 2017). The formulation of these classification methods involves the class-prior $\pi$, but $\pi$ is usually unknown in practice. If $\pi$ is poorly estimated, the classification accuracy of the state-of-the-art PU classification methods (du Plessis et al., 2014; 2015; Kiryo et al., 2017) could be degraded.

The mixture proportion estimation (MPE) is closely related to CPE (Blanchard et al., 2010; Scott, 2015). In the setting of MPE, there is a mixture distribution

$$F = (1 - \kappa^*)G + \kappa^* H, \tag{1}$$

where $H$ and $G$ are called component distributions. Given the samples randomly drawn from $F$ and $H$, respectively, MPE aims to estimate the maximum proportion $\kappa^* \in (0, 1)$ of $H$ in $F$. Thereby, if the maximum proportion $\kappa^*$ is identical to the class-prior $\pi$, the MPE methods can be employed to obtain $\pi$ by letting $P_{\mathrm{u}}$ and $P_{\mathrm{p}}$ be the mixture distribution $F$ and the component distribution $H$, respectively; otherwise, the MPE methods cannot be employed. To the best of our knowledge, most of state-of-the-art CPE methods (Blanchard et al., 2010; Liu & Tao, 2015; Scott, 2015; Ramaswamy et al., 2016; Jain et al., 2016) are based on MPE, which do not rely on assumptions that the data are drawn from a given parametric family of probability distributions (i.e., they are distributional-assumption-free methods).

To let these distributional-assumption-free methods can be used to identify class-prior $\pi$, $\kappa^*$ must be identical to the class-prior $\pi$. The *irreducibility* assumption (Blanchard et al., 2010) has been proposed to make them identical, which is employed by all these CPE methods implicitly or explicitly, to the best of our knowledge. It assumes that the support of the positive class-conditional distribution $P_{\mathrm{p}}$ is not contained in the support of the negative class-conditional distribution $P_{\mathrm{n}}$. However, it is strong and hard to be verified in PU learning, since $P_{\mathrm{n}}$ is a latent distribution, such that we do not have any prior knowledge about it. Additionally, since the applications of PU learning are diverse (Hsieh et al., 2019; Bekker & Davis, 2020), it is hard to guarantee that the support of $P_{\mathrm{p}}$ is not in the support of $P_{\mathrm{n}}$.

If the irreducibility assumption cannot be satisfied, the existing distributional-assumption-free CPE methods will suffer from an overestimation of $\pi$. For example, in Figure 1(a), we show both the unlabeled data distribution $P_{\mathrm{u}}$ and the component distribution $P_{\mathrm{p}}$. In Figure 1(b), we assume the latent negative class-conditional distribution $P_{\mathrm{n}}$ is fixed as shown in the green color, and the positive class-prior $\pi = 0.5$. In Figure 1(c), we show the existing distributional-assumption-free CPE methods will output the biased class-prior $0.7$. It is different from the ground truth $0.5$, since the support of $P_{\mathrm{p}}$ is contained in the support of $P_{\mathrm{n}}$. When the irreducibility assumption is not held, how to improve the estimations of distributional-assumption-free PU learning methods is challenging but useful.

Because the irreducibility assumption is impossible to check without making any assumption on $P_n$. Thereby, in this paper, we rethink those CPE methods and propose a novel method called *Regrouping CPE* (ReCPE) which improves the estimations of the current PU learning methods without irreducibility assumption. The main idea of our method is that, instead of estimating the maximum proportion of $P_p$ in $P_u$, we build a new CPE problem by creating a new auxiliary distribution $P_{p'}$ always guaranteeing the irreducibility assumption. Then we use the existing CPE method to obtain the maximum proportion of $P_{p'}$ in $P_u$, which is denoted by $\pi'$. We show that, with both theoretical analyses and experimental validations, when the irreducibility assumption holds, our ReCPE method does not affect the prediction of the existing estimators; when the irreducibility assumption does not hold, our method will help the current estimators have less estimation bias, which could improve the performances of PU classification tasks. For example, in Figure 1(d), we create a new class-conditional (auxiliary) distribution $P_{p'}$. By solving it, $\pi' = 0.51$. The estimation bias of the existing estimators will reduce to $\pi' - \pi = 0.01$ instead of $\kappa^* - \pi = 0.2$.

The rest of the paper is organized as follows. In Section 2, we review the irreducibility assumption and its variants. We discuss the difficulty of checking the assumptions. In Section 3, we provide the estimation biases of the existing consistent distributional-assumption-free CPE methods. Then we propose our method ReCPE, followed by theoretically analysis of its estimation bias and the implementation details. All the proofs are listed in Appendix A. The experimental validations are given in Section 4. Section 5 concludes the paper.

## 2    IRREDUCIBILITY OF CPE

In this section, we briefly review the assumptions used for existing distributional-assumption-free CPE estimators. Then we provide the estimation bias introduced by consistent distributional-assumption-free CPE methods when the assumptions do not hold.

**The irreducibility assumption.** Let $P_p$ and $P_u$ be probability measures (distributions) on a measurable space $(\mathcal{X}, \mathfrak{S})$, where $\mathcal{X}$ is the sample space, and $\mathfrak{S}$ is the $\sigma$-algebra. Let $\kappa^*$ be the maximum proportion of $P_p$ in $P_u$. To let $\kappa^*$ be identical to $\pi$, the irreducibility assumption was proposed by Blanchard et al. (2010).

**Definition 1** (Irreducibility). *$P_n$ and $P_p$ are said to satisfy the irreducibility assumption if $P_n$ is not a mixture containing $P_p$. That is, there does not exist a decomposition $P_n = (1 - \beta)Q + \beta P_p$, where $Q$ is a probability distribution on the measurable space $(\mathcal{X}, \mathfrak{S})$, and $0 < \beta \leq 1$.*

Equivalently, the assumption assumes the support of $P_p$ is hardly contained in the support of $P_n$. It means that with the selection of different sets $S$, the probability $P_n(S)$ can be arbitrarily close to $0$, and $P_p(S) > 0$. Suppose we can access the distributions $P_u$, $P_p$ and the set $\mathcal{C}$ containing all possible latent distributions, then the class-prior $\pi$ can be found as follows:

$$\pi = \kappa^* \triangleq \sup\{\alpha | P_u = (1 - \alpha)K + \alpha P_p, K \in \mathcal{C}\} = \inf_{S \in \mathfrak{S}, P_p(S) > 0} \frac{P_u(S)}{P_p(S)}. \tag{2}$$

To the best of our knowledge, all existing distributional-assumption-free CPE methods (Blanchard et al., 2010; Scott et al., 2013; Liu & Tao, 2015; Scott, 2015; Ramaswamy et al., 2016; Ivanov, 2019) are variants of estimating the maximum proportion $\kappa^*$ of $P_p$ in $P_u$. Many of them are statistically consistent estimators (Blanchard et al., 2010; Scott et al., 2013; Liu & Tao, 2015; Scott, 2015).

**The variants of the irreducibility.** Based on the irreducibility assumption, estimators can be designed with theoretical guarantees that they will converge to the class-prior $\pi$ (Blanchard et al., 2010). However, the convergence rate can be arbitrarily slow (Scott, 2015). The reason is that the irreducibility assumption implies the following fact (Blanchard et al., 2010; Scott et al., 2013)

$$\inf_{S \in \mathfrak{S}, P_p(S) > 0} \frac{P_n(S)}{P_p(S)} = 0, \tag{3}$$

i.e., the maximum proportion of $P_p$ in $P_n$ approaches to $0$. To obtain the class-prior $\pi$, it requires finding a sequence of the sets $S$ converging to the infimum, which empirically can be hard to find. Therefore, the convergence rate of the designed estimators based on Eq. (3) will be arbitrarily slow. To ensure a fixed rate of convergence, the *anchor set* assumption, a stronger variant of the irreducibility

assumption, has been proposed (Scott, 2015; Liu & Tao, 2015; Xia et al., 2019; 2020; Yao et al., 2020). It assumes that

$$\min_{S \in \mathfrak{S}, P_{\mathrm{p}}(S) > 0} \frac{P_{\mathrm{n}}(S)}{P_{\mathrm{p}}(S)} = 0, \tag{4}$$

i.e., there exists a set can achieve the minimum 0, which is called an anchor set. Another stronger variant is the *separability* assumption (Ramaswamy et al., 2016) which extends the anchor set assumption to a function space. It is proposed to bound the convergence rate of the method based on kernel-mean-matching (KMM) technique (Gretton et al., 2012).

## 3 REGROUPING FOR CPE (RECPE)

In this section, we propose a general method named regrouping for CPE (ReCPE). We discuss how to theoretically and empirically mitigate the overestimation problem of the class-prior $\pi$.

### 3.1 MOTIVATION

In general, it is impossible to verify the irreducibility assumption for CPE. To check the assumption, we need to make $P_{\mathrm{n}}$ itself to be observable and verify that whether the distribution $P_{\mathrm{n}}$ is a mixture containing the distribution $P_{\mathrm{p}}$, which obviously contradicts the setting of PU learning. However, in practice, the irreducibility assumption may not hold for many real-world problems, because the negative class is diverse (Hsieh et al., 2019; Bekker & Davis, 2020) in PU learning. If the assumption does not hold, $P_{\mathrm{n}}$ is said to be reducible to $P_{\mathrm{p}}$, and distributional-assumption-free CPE methods will introduce an *estimation bias*.

**Proposition 1.** *Let* $\beta^* = \inf_{S \in \mathfrak{S}, P_{\mathrm{p}}(S) > 0} \frac{P_{\mathrm{n}}(S)}{P_{\mathrm{p}}(S)}$ *be the maximum proportion of* $P_{\mathrm{p}}$ *in* $P_{\mathrm{n}}$, *given* $P_{\mathrm{u}} = (1 - \pi)P_{\mathrm{n}} + \pi P_{\mathrm{p}}$, *for* $0 < \pi \le 1$, *we have*

$$\kappa^* = \pi + (1 - \pi) \inf_{S \in \mathfrak{S}, P_{\mathrm{p}}(S) > 0} \frac{P_{\mathrm{n}}(S)}{P_{\mathrm{p}}(S)} = \pi + (1 - \pi)\beta^*. \tag{5}$$

According to Proposition 1, if the irreducibility assumption does not hold, then there exists $\beta > 0$. In this case, maximum proportion $\kappa^*$ can still be obtained, but it is different from $\pi$ but equal to $\pi + (1 - \pi)\beta^*$. In this case, if we directly employ existing distributional-assumption-free CPE methods, they could introduce an arbitrary estimation bias $(1 - \pi)\beta^*$ which depends on $P_{\mathrm{n}}$.

To reduce the estimation bias, we propose ReCPE. The process of regrouping is to change the original class-conditional distributions $P_{\mathrm{n}}$ and $P_{\mathrm{p}}$ into new class-conditional distributions $P_{\mathrm{n}'}$ and $P_{\mathrm{p}'}$ by transporting the probability mass of the set $A$ from the negative class to the positive class. After regrouping, new class-conditional distributions are guaranteed to satisfy the irreducibility assumption, and therefore, the new positive class-prior $\pi'$ can be identified by current CPE methods. To get the intuition, we provide a concrete example as follows.

Suppose that $P_{\mathrm{p}}$ is the uniform on $[\frac{1}{2}, 1]$, $P_{\mathrm{n}}$ is uniform on $[0, 1]$, and $\pi = \frac{1}{2}$. Then we have $P_{\mathrm{u}}$ such that it is uniform on $[0, \frac{1}{2})$ and $[\frac{1}{2}, 1]$, respectively. Specifically, the probabilities are

$$P_{\mathrm{u}}\left([0, \frac{1}{2})\right) = \frac{1}{4}, \quad P_{\mathrm{u}}\left([\frac{1}{2}, 1]\right) = \frac{3}{4}.$$

In this case, by Eq. 5, the maximum proportion of $P_{\mathrm{p}}$ in $P_{\mathrm{u}}$ is $\kappa^* = \frac{3}{4}$. Let $\rho > 0$ be a small constant, and let $A = (1 - \rho, 1]$. In this case, the mass of $A$ in $P_{\mathrm{u}}$ from $P_{\mathrm{n}}$ is $\pi P_{\mathrm{n}}(A) = \frac{\rho}{2}$. After transporting the mass $\frac{\rho}{2}$ from $P_{\mathrm{n}}$ to $P_{\mathrm{p}}$, we have a new positive class-prior $\pi'$ and a new class-conditional $P_{\mathrm{p}'}$ which is uniform on $[\frac{1}{2}, 1 - \rho)$ and $[1 - \rho, 1]$, respectively. Specifically,

$$\pi' = \frac{1 + \rho}{2}, \quad P_{\mathrm{p}'}\left([\frac{1}{2}, 1 - \rho)\right) = \frac{1 - 2\rho}{1 + \rho}, \quad P_{\mathrm{p}'}\left([1 - \rho, 1]\right) = \frac{3\rho}{1 + \rho}.$$

The left equation above shows that the new class-prior $\pi'$ is dependent on $\rho$ or the size of $A$. By controlling set $A$ or $\rho$ to be small, $\pi'$ can be as close to $\pi$ as possible. This is the intuition of how regrouping works.

---

**Algorithm 1** ReCPE

> **Input:** An unlabeled sample $S_{\mathrm{u}}$ i.i.d. drawn from $P_{\mathrm{u}}$, a positive sample $S_{\mathrm{p}}$ i.i.d. drawn from $P_{\mathrm{p}}$, and the percentage $p$ of the sample needed to copy from $S_{\mathrm{u}}$ to $S_{\mathrm{p}}$.
> 1: Train a binary classifier $h$ with the unlabeled sample $S_{\mathrm{u}}$ and positive sample $S_{\mathrm{p}}$ by treating $S_{\mathrm{u}}$ as a negative sample;
> 2: Assign each example $x \in S_{\mathrm{u}}$ with the negative class-posterior probability $P(Y = -1|X = x)$ predicted by the trained classifier $h$;
> 3: Obtain $S_{\mathrm{p'}}$ by copying $p \times |S_{\mathrm{u}}|$ examples with the smallest negative class-posterior probability $P(Y = -1|X = x)$ from $S_{\mathrm{u}}$ to $S_{\mathrm{p}}$;
> 4: Estimate the class-prior $\pi'$ by employing an algorithm based on Eq. (5) with inputs $S_{\mathrm{u}}$ and $S_{\mathrm{p'}}$.
> **Output:** The estimated new class-prior $\hat{\pi}'$.

---

### 3.2 PRACTICAL IMPLEMENTATION

In practice, we have to implement the aforementioned idea of regrouping based on positive sample $S_{\mathrm{p}}$ and unlabeled sample $S_{\mathrm{u}}$. Since the negative sample is unavailable, we cannot "cut and paste" any example from negative class to positive sample $S_{\mathrm{p}}$; instead, we can "copy and past" some unlabeled examples to $S_{\mathrm{p}}$. When doing so, we should select a small set of samples $\hat{A}^*$ which look the most similar to the positive class and dissimilar to the negative class, which could encourage the difference between the original $P_{\mathrm{n}}$, $P_{\mathrm{p}}$, and $\rho$ and $\pi$, $P_{\mathrm{p'}}$, and $\pi'$ to be small. This is why $\hat{A}^* = (1 - \rho, 1]$ was selected in the above intuitive example, i.e., $\hat{A}^*$ belongs geometrically and visually to the positive class with the highest confidence among all subsets of $[0, 1]$ of size $\rho$.

A hyper-parameter $p \in (0, 1)$ is introduced to control the size of set $\hat{A}^*$, theoretically, we prefer the set $\hat{A}^*$ to have a small size. Empirically, $p$ cannot be so small: the existing estimators are insensitive to tiny modifications (they are designed to be robust in such a way, in order to be good estimators). For example, the difference between the estimated class-priors by employing samples $S_{\mathrm{u}}$ and $S_{\mathrm{p}}$ and the one by employing samples $S_{\mathrm{u}}$ and $S_{\mathrm{p'}}$ can be hardly observed if $S_{\mathrm{p}}$ and $S_{\mathrm{p'}}$ only differ from in one or two points. Specifically, $p = 10\%$ is selected for the experiments on all datasets, which leads to a significant improvement of the estimation accuracy. The details on the selection of the hyper-parameter value will be explained in Section 4.1. The algorithm is summarized in Algorithm 1.

There are two fundamental concerns for copying $\hat{A}^*$ to $S_{\mathrm{p}}$. 1). When we have irreducibility, might regrouping make $\hat{\pi}'$ be a worse approximation? 2). When we lack irreducibility, must regrouping make $\hat{\pi}'$ be a better approximation? While these concerns will be formally clarified later, we give here intuitive implications of regrouping.

1). If we have irreducibility, the $P_{\mathrm{n}}(\hat{A}^*)$ should be rather small (if not zero), and $\hat{A}^*$ should be drawn from the positive component $P_{\mathrm{p}}$ of the mixture $P_{\mathrm{u}}$. In this case, regrouping will generally have small influence to $P_{\mathrm{p}}$. Hence, it will not make $\hat{\pi}'$ worse.

2). If we lack irreducibility, $\hat{A}^*$ may be drawn from either $P_{\mathrm{p}}$ or $P_{\mathrm{n}}$. By regrouping, $\hat{A}^*$ becomes present in $S_{\mathrm{p'}}$, which encourages the probability of the set $\hat{A}^*$ in $P_{\mathrm{p'}}$ to be large. This will modify $P_{\mathrm{p}}$ as we expected towards irreducibility. As a consequence, regrouping will make $\hat{\pi}'$ better.

### 3.3 THEORETICAL JUSTIFICATION

In the regrouping approach described above, the auxiliary class-conditional distribution $P_{\mathrm{p'}}$ and $P_{\mathrm{n'}}$ are created by regrouping a small set $A$ from $P_{\mathrm{p}}$ and $P_{\mathrm{n}}$. Here, we analyze the properties of regrouping and theoretically justify it.

**A formal definition of regrouping** In order to analyze the properties, we need to formally define how to split, transport, and regroup a set $A$ (or the mass of $A$).

**Definition 2.** *Let $M$ be a probability measure on a measurable space $(\mathcal{X}, \mathfrak{S})$. Given a set $A \in \mathfrak{S}$, we define a measure $M^A$ on the $\sigma$-algebra $\mathfrak{S}$ as follows:*

$$\forall S \in \mathfrak{S}, M^A(S) = M(S \cap A). \tag{6}$$

It is easy to see that given two measures $M^A$ and $M^{A^c}$ obtained according to Definition 2, where $A^c = \mathcal{X} \setminus A$, then $M^A$ and $M^{A^c}$ have the following property.

**Lemma 1.** *Let $M$ be a probability measure over a measurable space $(\mathcal{X}, \mathfrak{S})$. For any set $A \in \mathfrak{S}$, we have $M^A + M^{A^c} = M$.*

Now, we introduce the theory of regrouping. Fixing a set $A \in \mathfrak{S}$, we split $P_n$ as $P_n^{A^c}$ and $P_n^A$, transport $P_n^A$ to $P_p$ to regroup them together, i.e.,

$$P_u = (1-\pi)P_n + \pi P_p = (1-\pi)\underbrace{(P_n^A + P_n^{A^c})}_{\text{split into two}} + \pi P_p = (1-\pi)P_n^{A^c} + \underbrace{((1-\pi)P_n^A + \pi P_p)}_{\text{regroup as one}}.$$

Finally, we can rewrite the unlabeled data distribution $P_u$ as a mixture of two new class-conditional distributions $P_{n'}$ and $P_{p'}$ defined in Theorem 1 by normalization.

**Theorem 1.** *Let $P_u = (1-\pi)P_n + \pi P_p$. Let $A \subset \text{support}(P_u)$. By regrouping $P_n^A$ to $P_p$, $P_u$ can be written as a mixture, i.e., $P_u = (1-\pi')P_{n'} + \pi' P_{p'}$, where*

$$\pi' = \pi + (1-\pi)P_n(A), \tag{7}$$

$$P_{n'} = \frac{P_n^{A^c}}{P_n(A^c)}, \quad P_{p'} = \frac{(1-\pi)P_n^A + \pi P_p}{(1-\pi)P_n(A) + \pi}, \tag{8}$$

*and $P_{n'}$ and $P_{p'}$ satisfy the anchor set assumption.*

When class-conditional distributions $P_n$ and $P_p$ do not satisfy the irreducibility assumption, $\pi$ cannot be obtained by using CPE methods based on MPE, which will lead to an estimation bias discussed before. However, Theorem 1 shows that the new proportion $\pi'$ is always identifiable as $P_{n'}$ and $P_{p'}$ always satisfy the anchor set assumption. Thus, after regrouping, $\pi'$ is identifiable and can be estimated by the existing CPE methods.

**Bias reduction** According to Theorem 1, to make $\pi'$ closer to $\pi$, we expect to find the set $A$ looks most dissimilar to the negative class, i.e., $P_n(A)$ is small.

**Theorem 2.** *Let $P_{p'}$ and $P_{n'}$ be obtained by regrouping a set $A^* := \arg\min_{A \in \mathfrak{S}} \frac{P_n(A)}{P_p(A)}$[1] from $P_p$ and $P_n$. 1). If $P_n$ and $P_p$ satisfy the irreducibility assumption, then $\pi' = \pi$; 2). if $P_n$ and $P_p$ dissatisfy the irreducibility assumption, then $\pi < \pi' < \pi + (1-\pi)\beta^* = \kappa^*$.*

Theorem 2 shows how to properly select a set used for regrouping to make $\pi'$ a good approximation of $\pi$. Specifically, once $A^*$ is selected for regrouping, if $P_n$ and $P_p$ satisfy the irreducibility assumption, the new estimation $\pi'$ will be identical to $\pi$; if $P_n$ and $P_p$ dissatisfy the irreducibility assumption, $\pi'$ obtained by employing the distributions $P_u$ and $P_{p'}$ will contain a smaller estimation bias compared to $\kappa^*$ obtained by employing the distributions $P_u$ and $P_p$.

**Convergence analysis** For completeness, we illustrate the convergence property of ReCPE, which is presented by employing the estimator proposed by Blanchard et al. (2010). Let $S_u$, $S_p$ and $S_{p'}$ be the samples i.i.d. drawn from $P_u$, $P_p$ and $P_{p'}$, respectively. Let $A$ be the set used for regrouping. Let $h : \mathcal{X} \to \mathbb{R}, h \in \mathcal{H}$, be a function that predicts 1 for all elements in the set $A$ and 0 otherwise, where $\mathcal{H}$ denotes a *hypothesis space*. Let $|S|$ denote the cardinality of a set S. Let $\mathbb{1}_{\{h(x)=1\}}$ be an indicator function which returns 1 if $h(x)$ predicts 1 and 0 otherwise. Then $P_u(A)$ can be expressed as $\int_{x \in \mathcal{X}} p_p(x) \mathbb{1}_{\{h(x)=1\}} \mathrm{d}x$, where $p_p$ is the density function of the distribution $P_u$. Let $\hat{P}_u(A)$ be the empirical version of $P_u(A)$, i.e., $\hat{P}_u(A) = \frac{1}{|S_u|} \sum_{x \in \mathcal{X}} \mathbb{1}_{\{h(x)=1\}}$. Similarly, let $\hat{P}_{p'}(A)$ be the empirical version of $P_{p'}(A)$. Let the error $\epsilon_{\delta, \mathcal{H}}(S_u)$ denote the difference between $P_u(A)$ and $\hat{P}_u(A)$ obtained by exploiting the *empirical Rademacher complexity* (Mohri et al., 2018). Similarly, let $\epsilon_{\delta, \mathcal{H}}(S_{p'})$ denote the difference between $P_{p'}(A)$ and $\hat{P}_{p'}(A)$. We have the following theorem.

**Theorem 3.** *Let $P_u = (1-\pi)P_n + \pi P_p$. By selecting a set $A$ and regrouping $P_n^A$ to $P_p$. Then, with probability $1 - 2\delta$, the estimated class-prior $\hat{\pi}'$ based on solving $\inf_{S \in \mathfrak{S}, \hat{P}_{p'}(S) > 0} \frac{\hat{P}_u(S)}{\hat{P}_{p'}(S)}$ satisfies*

$$|\hat{\pi}' - \pi| \leq \frac{\epsilon_{\delta, \mathcal{H}}(S_{p'})}{\hat{P}_{p'}(A) + \epsilon_{\delta, \mathcal{H}}(S_{p'})} + \frac{\epsilon_{\delta, \mathcal{H}}(S_u)}{\hat{P}_{p'}(A) + \epsilon_{\delta, \mathcal{H}}(S_{p'})} + (1-\pi)P_n(A), \tag{9}$$

---

[1]We have defined that the fraction tends to infinite if its numerator is larger than 0 and its denominator is 0. Additionally, the infimum may not always exist, if it does not exist, we could use a sequence of sets that converges to the infimum value, but the convergence rate can be arbitrarily slow (Scott, 2015).

*where $\epsilon_{\delta,\mathcal{H}}(S) \triangleq 2\hat{\mathfrak{R}}_S(\mathcal{H}) + 3\sqrt{\frac{\log\frac{4}{\delta}}{2|S|}}$, and $\hat{\mathfrak{R}}_S(\mathcal{H})$ is the empirical Rademacher complexity of $\mathcal{H}$.*

To make $\epsilon_{\delta,\mathcal{H}}(S)$ converge to $0$ with the increasing of the sample size of $S$, a *universal approximation* assumption has been proposed by Scott (2015) to ensure that the hypothesis space is large enough to represent a wide variety of interesting functions. Under the assumption, Scott (2015) proved that, with increasing of the size of samples $S_{\mathrm{u}}$ and $S_{\mathrm{p}'}$, the error between $P_{\mathrm{u}}(A)$ and $\hat{P}_{\mathrm{u}}(A)$ will converge to $0$ at a rate $\mathcal{O}\left(\sqrt{\frac{\log|S_{\mathrm{u}}|}{|S_{\mathrm{u}}|}}\right)$, similarly to $P_{\mathrm{p}'}(A)$ and $\hat{P}_{\mathrm{p}'}(A)$. Since the empirical Rademacher complexity $\hat{\mathfrak{R}}_X(\mathcal{H})$ of a hypothesis space $\mathcal{H}$ can be upper-bounded by its VC-dimension (Mohri et al., 2018), the both errors based on the empirical Rademacher complexity will also converge to zero with increasing of the sample size. Consequently, the estimation $\hat{\pi}' = \frac{\hat{P}_{\mathrm{u}}(A)}{\hat{P}_{\mathrm{p}'}(A)}$ will converge to $\pi' = \frac{P_{\mathrm{u}}(A)}{P_{\mathrm{p}'}(A)} = \pi + (1-\pi)P_{\mathrm{n}}(A)$ at a rate $\mathcal{O}\left(\sqrt{\frac{\log(\min(|S_{\mathrm{u}}|,|S_{\mathrm{p}'}|))}{\min(|S_{\mathrm{u}}|,|S_{\mathrm{p}'}|)}}\right)$.

**Computationally efficient identification of $A^*$**  The following theorem presents how to identify $A^*$ with $P_{\mathrm{u}}$ and $P_{\mathrm{p}}$. Let us define another auxiliary distribution $q(X,C)$, where $C \in \{0,1\}$ is the positive-vs-unlabeled label i.e., a class label distinguishing between the positive component and the whole mixture. Specifically, priors are $q(C=1) := \frac{\pi}{1-\pi}$ and $q(C=0) := \frac{1}{1-\pi}$; conditional densities are $q(X|C=1) := P_{\mathrm{p}}$ and $q(X|C=0) := P_{\mathrm{u}}$; class-posterior probabilities are $q(C=0|X)$ and $q(C=1|X)$. We have the following theorem.

**Theorem 4.** *Let $p_{\mathrm{u}}$ and $p_{\mathrm{p}}$ be density functions of $P_{\mathrm{u}}$ and $P_{\mathrm{p}}$, respectively. Let $q = P(C=0)p_{\mathrm{u}} + P(C=1)p_{\mathrm{p}}$. Let $\mathbb{1}_A : \mathcal{X} \to \{0,1\}$ be the identity function which outputs $1$ if $x \in \mathcal{X}$ is in the set $A$, and $0$ otherwise. Then the set $A^* = \arg\min_{A \in \mathfrak{S}} \frac{\mathbb{E}_{x \sim q(X)}[\mathbb{1}_A(X=x)q(C=0|X=x)]}{\mathbb{E}_{x \sim q(X)}[\mathbb{1}_A(X=x)q(C=1|X=x)]}$.*

For the above optimization, its objective function has two expectations over $q$, which can have the "exact" empirical solution obtained by replacing expectations with empirical averages: $A^* = \arg\min_{A \subset S} \frac{\sum_{x \in A} q(C=0|X=x)}{\sum_{x \in A} q(C=1|X=x)}$.

**Approximation of $P_{\mathrm{p}'}$ with a surrogate**  As we do not have examples drawn from $P_{\mathrm{n}}$, it is hard to create $P_{\mathrm{p}'}$, let alone sample from it. We approximate $P_{\mathrm{p}'}$ by using $P_{\tilde{\mathrm{p}}'} = \frac{P_{\mathrm{u}}^A + P_{\mathrm{p}}}{P_{\mathrm{u}}(A)+1}$. The following proposition shows that when $P_{\mathrm{u}}(A)$ is small, $P_{\tilde{\mathrm{p}}'}$ is almost identical to $P_{\mathrm{p}'}$.

**Proposition 2.** *Let $P_{\tilde{\mathrm{p}}'} = \frac{P_{\mathrm{u}}^A + P_{\mathrm{p}}}{P_{\mathrm{u}}(A)+1}$ and $P_{\mathrm{u}}(A) < \epsilon$. $\forall \epsilon > 0$ and $S \in \mathfrak{S}$, $|P_{\mathrm{p}'}(S) - P_{\tilde{\mathrm{p}}'}(S)| \leq \mathcal{O}(\epsilon)$.*

Since the gap has the same order as $P_{\mathrm{u}}(A)$ uniformly over $S$, it is guaranteed that whenever $P_{\mathrm{u}}(A)$ is small, the gap is also small. Practically, we can control the parameter $\epsilon$ in the above proposition to be small. Specifically, using a small value of the hyper-parameter $p$ in Algorithm 1 will lead to the set $A$ in Theorem 1 to be small, as well as $P_{\mathrm{u}}(A)$. As a consequence, the practical implementation of regrouping is a good approximation of the theory of regrouping as we expected. By now, we have analyzed all of the properties of regrouping and theoretically justified all of the points in its design.

## 4 EXPERIMENTS

We run experiments on 2 synthetic datasets and 9 real word datasets[2]. The objectives of employing synthetic datasets are to validate whether the proposed regrouping CPE method reduces the estimation error of the consistent distributional-assumption-free CPE method on the dataset satisfying the irreducibility assumption and does not influence the prediction of the CPE method on the dataset dissatisfying the irreducibility assumption. The hyper-parameter $p$ is also selected from the synthetic datasets. The real-world datasets are used to illustrate the effectiveness of our methods. Although we have introduced a hyper-parameter $p$ and used approximations in the implementation, empirical results on all synthetic and real-world datasets consistently show the superiority of ReCPE.

To have a rigorous performance evaluation, for each dataset, $6 \times 3 \times 10$ experiments are conducted via random sampling. Specifically, we select $\{0.25, 0.5, 0.75\}$ fraction of positive examples to be the

---

[2]The real word datasets are downloaded from the UCL machine learning database. Multi-class datasets are used as binary datasets by either grouping or ignoring classes.

sample of the positive distribution $P_p$. We let the rest of the examples be the sample of the unlabeled distribution $P_u$. In such a way, 3 pairs of empirical positive and unlabeled distributions are generated. Then, we create other 3 pairs of distributions by flipping the labels of all instances in the original datasets. For each pair of distributions, we randomly draw positive and unlabeled samples with sizes of 800, 1600, and 3200, respectively, which are used as input data. Note that, the positive and unlabeled samples have the same size as did in Ramaswamy et al. (2016). For each sample size, 10 repeated experiments are carried out with random sampling. For all experiments, we employ a neural network [3] with 2 hidden layers. Each hidden layer contains 50 hidden units. The batch normalization (Ioffe & Szegedy, 2015) is also employed. The stochastic gradient descent optimizer is used with the batch size 50. The network is trained for 350 epochs with a learning rate 0.01 and momentum 0. The weight decay is set to $1e - 5$. The model with the best validation accuracy is used to estimate the positive class-posterior probability $P(Y = 1|X = x)$. We sample the validation set with 20% of the training data size.

### 4.1 Experiments on Synthetic Datasets

We create two datasets with one satisfying the irreducibility assumption while the other not. The dataset satisfying the irreducibility assumption is created by sampling from 2 different 10-dimensional Gaussian distributions as the component distributions. One of the distributions has zero means and a unit covariance matrix. Another one has unit means and unit covariance matrix. The dataset dissatisfying the irreducibility assumption is also created by drawing examples from 2 different 10-dimensional Gaussian distributions. One of the distributions has zero means and unit covariance matrix. Another one has unit means and covariance matrix. Then we remove all the data points with $P(Y = 1|X) \geq 0.98$ or $P(Y = 1|X) \leq 0.02$. For simplicity, in Figure 2, we name two datasets irreducible data and reducible data, respectively.

To validate the correctness of our method and to select a suitable value of the hyper-parameter $p$, we carry out two experiments. The consistent CPE method KM2 is used as the baseline, which is compared to our method ReKM2, i.e., regrouping version of the KM2. Firstly, we compare the magnitude differences between $\hat{\pi}$ and $\hat{\pi}'$ (i.e., $\hat{\pi} - \hat{\pi}'$) with the different fractions of points to be copied from the mixture sample to the component sample, which is illustrated in Figure 2(a). Then we compare differences of the absolute error (i.e., $|\hat{\pi} - \pi| - |\hat{\pi}' - \pi|$) between the baseline and our method with the increasing of the copy fractions. Note that each point in Figure 2 is obtained by averaging over $6 \times 3 \times 10$ experiments.

Figure 2(a) validates the correctness of our Theorem 2 and Eq. (7). Theorem 2 states that, by properly selecting the set $A$, on the dataset dissatisfying the irreducibility assumption (reducible data), $\pi'$ should be smaller than the maximum proportion $\kappa^*$; on the dataset satisfying the irreducibility assumption (irreducible data), $\pi'$ should be close to $\pi$. Figure 2(a) perfectly matches this statement. It shows that, on the reducible data,

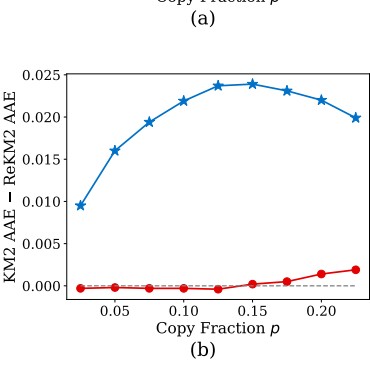

Figure 2: Experiments of the hyper-parameter selection on synthetic datasets. With increasing of the copy fraction $p$, (a) average estimation differences between KM2 and Regrouping KM2 (ReKM2) and (b) average differences of the absolute error between KM2 and Regrouping KM2 (ReKM2).

the values of $\hat{\pi}'$ are continuously smaller than $\hat{\pi}$ with the copy fraction $\leq 22.5\%$; on the irreducible data, $\hat{\pi}'$ and $\hat{\pi}$ have the similar values until the copy fraction $\geq 17.5\%$. According to Eq. (7), the positive bias of our estimator should become larger with the increase of $P_n(A)$. This fact is reflected by the differences of $\hat{\pi} - \hat{\pi}'$ become smaller on both datasets when the copy fraction $> 15\%$.

Figure 2(b) illustrates the average differences of absolute error between the baseline and the proposed method. On the reducible data, our method continuously outperforms the baseline with the copy

---

[3]We employ the neural network because it has a high approximation capability (Csáji et al., 2001).

| | AM | ReAM | DPL | ReDPL | EN | ReEN | KM1 | ReKM1 | KM2 | ReKM2 | ROC | ReROC | RPG | ReRPG |
|---|---|---|---|---|---|---|---|---|---|---|---|---|---|---|
| adult (800) | **0.127** | 0.13 | 0.122 | **0.108**∗ | 0.316 | **0.295** | 0.255 | **0.132** | 0.164 | **0.153** | 0.176 | **0.153** | 0.135 | **0.134** |
| adult (1600) | **0.122** | 0.124 | **0.089**∗ | 0.089∗ | 0.31 | **0.29** | 0.131 | **0.091** | **0.12** | 0.13 | 0.121 | **0.095** | **0.123** | 0.137 |
| adult (3200) | 0.105 | **0.086** | **0.054** | 0.057 | 0.297 | **0.279** | 0.054 | **0.04**∗ | **0.082** | 0.089 | 0.089 | **0.067** | **0.114** | 0.128 |
| avila (800) | 0.168 | **0.152** | **0.129** | 0.147 | 0.447 | **0.422** | 0.105 | **0.075**∗ | 0.104 | **0.081** | 0.263 | **0.228** | 0.119 | **0.111** |
| avila (1600) | 0.165 | **0.132** | 0.104 | **0.084** | 0.439 | **0.418** | 0.086 | **0.076**∗ | 0.108 | **0.092** | 0.191 | **0.16** | 0.123 | **0.121** |
| avila (3200) | 0.156 | **0.133** | **0.05**∗ | 0.061 | 0.436 | **0.42** | 0.092 | **0.078** | 0.112 | **0.092** |  | **0.095** | **0.121** | 0.122 |
| bank (800) | **0.135** | 0.158 | **0.116**∗ | 0.132 | 0.282 | **0.264** | 0.356 | **0.216** | 0.266 | **0.238** | 0.163 | **0.15** | **0.163** | 0.185 |
| bank (1600) | **0.117** | 0.167 | **0.087**∗ | 0.105 | 0.262 | **0.244** | 0.178 | **0.128** | 0.203 | **0.198** | 0.129 | **0.118** | **0.157** | 0.167 |
| bank (3200) | **0.104** | 0.127 | **0.073**∗ | 0.091 | 0.248 | **0.237** | 0.124 | **0.09** | **0.15** | 0.16 | **0.093** | 0.106 | **0.159** | 0.18 |
| card (800) | 0.131 | **0.127**∗ | 0.174 | **0.161** | 0.465 | **0.444** | 0.293 | **0.176** | 0.203 | **0.158** | 0.247 | **0.233** | 0.177 | **0.155** |
| card (1600) | 0.173 | **0.14** | 0.14 | 0.14 | 0.459 | **0.437** | 0.19 | **0.135** | 0.159 | **0.129** | 0.194 | **0.163** | 0.126 | **0.115**∗ |
| card (3200) | 0.164 | **0.134** | 0.127 | **0.12** | 0.455 | **0.435** | 0.161 | **0.113** | 0.142 | **0.122** | 0.159 | **0.152** | 0.11 | **0.108**∗ |
| covtype (800) | 0.16 | **0.123** | 0.155 | **0.151** | 0.367 | **0.343** | 0.157 | **0.142** | **0.122** | 0.13 | 0.291 | **0.258** | 0.116 | **0.105**∗ |
| covtype (1600) | 0.12 | **0.1**∗ | 0.132 | **0.109** | 0.364 | **0.339** | 0.116 | **0.113** | **0.121** | 0.123 | 0.199 | **0.161** | 0.109 | **0.108** |
| covtype (3200) | 0.128 | **0.09** | 0.093 | **0.083**∗ | 0.354 | **0.334** | **0.097** | 0.109 | **0.124** | 0.128 | 0.157 | **0.113** | 0.109 | **0.107** |
| egg (800) | 0.153 | **0.106**∗ | **0.218** | 0.225 | 0.505 | 0.505 | **0.173** | 0.264 | **0.119** | 0.131 | 0.476 | **0.396** | 0.171 | **0.124** |
| egg (1600) | 0.137 | **0.12** | **0.121** | 0.142 | **0.486** | 0.489 | 0.234 | **0.214** | 0.116 | **0.108**∗ | 0.315 | **0.238** | 0.151 | **0.114** |
| egg (3200) | 0.126 | **0.113** | **0.057**∗ | 0.073 | **0.485** | 0.489 | 0.26 | **0.193** | 0.134 | **0.113** | 0.163 | **0.139** | 0.142 | **0.102** |
| magic04 (800) | 0.099 | **0.077** | 0.072 | **0.071** | 0.312 | **0.296** | 0.111 | **0.1** | 0.071 | **0.064** | 0.141 | **0.124** | 0.055 | **0.054**∗ |
| magic04 (1600) | 0.071 | **0.056** | 0.044 | **0.043**∗ | 0.292 | **0.274** | 0.084 | **0.072** | 0.079 | **0.065** | 0.1 | **0.073** | 0.058 | **0.052** |
| magic04 (3200) | 0.069 | **0.054** | **0.035**∗ | 0.036 | 0.274 | **0.258** | 0.07 | **0.047** | 0.085 | **0.063** | 0.065 | **0.047** | 0.054 | **0.052** |
| robot (800) | **0.053** | 0.062 | 0.049 | **0.047**∗ | 0.19 | **0.187** | 0.232 | **0.215** | **0.111** | 0.114 | **0.119** | 0.144 | **0.077** | 0.084 |
| robot (1600) | 0.053 | **0.038**∗ | 0.087 | **0.054** | 0.139 | **0.132** | 0.15 | **0.141** | **0.098** | 0.099 | 0.08 | **0.075** | **0.076** | 0.079 |
| robot (3200) | 0.052 | **0.039**∗ | 0.156 | **0.119** | 0.091 | **0.085** | 0.079 | **0.077** | 0.084 | 0.084 | 0.063 | **0.043** | **0.06** | 0.066 |
| shuttle (800) | 0.083 | **0.031** | **0.016**∗ | 0.02 | 0.041 | **0.035** | **0.058** | 0.083 | **0.035** | 0.065 | **0.042** | 0.047 | **0.035** | 0.051 |
| shuttle (1600) | 0.09 | **0.045** | **0.011**∗ | 0.018 | 0.04 | **0.034** | **0.048** | 0.079 | **0.024** | 0.05 | **0.029** | 0.043 | **0.026** | 0.039 |
| shuttle (3200) | 0.076 | **0.028** | **0.012**∗ | 0.021 | 0.043 | **0.038** | **0.046** | 0.07 | **0.018** | 0.03 | **0.038** | 0.045 | **0.028** | 0.042 |
| average | 0.116 | **0.1** | 0.094 | **0.092**∗ | 0.311 | **0.297** | 0.146 | **0.121** | 0.117 | **0.111** | 0.157 | **0.136** | 0.106 | **0.105** |

Table 1: Absolute estimation errors on real-world datasets. The first column provides the names of the datasets and sample size. We bold the smaller average estimation errors by comparing each baseline method with its regrouped version. The smallest average estimation error among all methods for each row is highlighted with ∗. The last row is obtained by averaging the results of all experiments. Variances and the results of Wilcoxon signed-rank test are reported in Appendix B. The proposed Regrouping methods are significantly better than most of the baselines.

fraction $\leq 22.5\%$. However, the differences of average absolute error start to decrease with the copy fraction $> 15\%$. On the irreducible data, the differences of average absolute error are close to zero until the copy fraction $> 15\%$.

By observing Figure 2, we found the prediction of the KM2 estimator will not change much if the copy fraction $p$ is too small. For example, the difference between the estimated mixture proportion by employing samples $S_{\mathrm{u}}$ and $X_P$ and the ones by employing samples $S_{\mathrm{u}}$ and $X_P'$ can be hardly observed if $X_P$ and $X_P'$ only differ from in one or two points. For simplicity and consistency, we select hyper-parameter $p$ to be $10\%$ for all the following experiments.

## 4.2 Experiments on Real-world Datasets

We illustrate the absolute estimation errors of different estimators on the real-world datasets. Totally, 7 baseline methods are used in the experiments, which are AlphaMax (AM) (Jain et al., 2016), DEDPUL (DPL) (Ivanov, 2019), Elkan-Noto (EN) (Elkan & Noto, 2008), KM1, KM2 (Ramaswamy et al., 2016), ROC (Scott, 2015), and Rankpruning (RPG) (Northcutt et al., 2017). By using our method, the regrouped version of them are implemented, which are called ReAM, ReDPL, ReEN, ReKM1, ReKM2, ReROC, and ReRPG. In Table 1, we compare the absolute estimation errors of each baseline with those of its regrouped version on different datasets with different sample lengths. Each number in Table 1 is the average over $6 \times 10$ experiments.

Table 1 reflects the effectiveness of our regrouping CPE method. Overall, by using our method, the estimation accuracy is increased for most of the popular CPE methods among most of the datasets with different sample lengths. By observing the last row, the regrouped version of the estimators has much smaller average estimation errors except DPL, KM2, and RPG. On the real-world datasets, Regrouping AlphaMax (ReAM) has the smallest average estimation error among all methods.

## 5 Conclusion

In this paper, we investigate how to reduce the estimation bias of the distributional-assumption-free CPE method without irreducibility assumption for PU learning. We have proposed regrouping CPE which can be employed on top of most existing CPE methods. We have also theoretically analyzed the estimation bias of ReCPE. Empirically, it improves all popular CPE methods on various datasets. One future work will focus on how to generate a sample from $P_{\mathrm{p}'}$ instead of using an approximation.

## REPRODUCIBILITY STATEMENT

For theoretical results, we have clearly explained any assumptions. A complete proof of the claims can be founded in the appendix. We have also included an anonymous source code in our supplementary material. For any datasets used in the experiments, a complete description of the data processing steps is provided In Section 4.

## ACKNOWLEDGMENTS

TL was partially supported by Australian Research Council Projects DP180103424, DE-190101473, IC-190100031, and DP-220102121. BH was supported by the RGC Early Career Scheme No. 22200720 and NSFC Young Scientists Fund No. 62006202. MG is supported by ARC DE210101624. GN and MS were supported by JST AIP Acceleration Research Grant Number JPMJCR20U3, Japan. MS was also supported by the Institute for AI and Beyond, UTokyo.

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
