# OpenReview forum: "Rethinking Class-Prior Estimation for Positive-Unlabeled Learning"
_ICLR.cc/2022/Conference — ICLR 2022 Poster_

### Official Review · Reviewer_6Jj8 · 2021-10-30

**Correctness:** 4
**Technical Novelty And Significance:** 3
**Empirical Novelty And Significance:** 3
**Recommendation:** 6
**Confidence:** 4

**Main Review:**

Strengths:

1. This paper introduces a particular situation in which the irreducibility assumption cannot be satisfied, which has not been studied (even considered) in previous PU learning literature.

2. The idea of "regrouping" is novel. The authors give a very simple and intuitive example in Subsection 3.1 as the motivation of the proposed method, which is very interesting and helps the readability of this paper. In Subsection 3.3, some theoretical results are introduced and these results can help the readers to understand the proposed "regrouping' idea.

3. The experimental results on synthetic data validate the correctness of the theoretical part and show the effectiveness of the proposed method in "reducibility" setting.

Weaknesses:

1. The experiments on real-world data use same  hyper-parameter $p$ across all datasets. Considering that this paper actually aims to address the hyper-parameter (class-prior $\pi$) selection problem and the proposed method introduces another hyper-parameter $p$ (which seems to have significant impact to the performance), the parameter sensitivity analysis is needed in experiment.

2. More related work about "irreducibility" can be introduced before Section 3. I don't know is there any solution for the irreducibility problem in mixture proportion estimation after reading the whole paper.

Other comments:

1. Is the indicator function really needed in "Convergence analysis" part? It seems that $\mathbb{1}_{\{h(x)=1\}}$ is equivalent to $h(x)$.

2. Can you explain that why "In this case, regrouping will generally not change $P_p$ but only increase number of examples in $P_p$".

3. Typos: "diffferent" --> "defferent", "the trained classifier a h" --> "the trained classifier h".

**Summary Of The Paper:**

This paper addresses the problem of class-prior estimation (CPE) in positive-unlabeled (PU) learning. The authors consider that the existing methods usually fail if the data distribution dissatisfies the irreducibility assumption. To address this problem, the authors introduce a method named Regroping CPE (ReCPE) which tries to transform the original data distribution to an auxiliary data distribution, such that the produced distribution always guaranteeing the irreducibility assumption.

A practice implementation is proposed by firstly training a binary classifier and then picking some samples whose outputs are mostly dissimilar to the negative examples as the pseudo positive samples. The proposed methods can be considered as a pre-processing method and used together with any CPE method. The experiments on synthetic data show the intuition of the proposed ReCPE and the results on real-world data show the effectiveness of the proposed method when it is combined with other CPE methods.

**Summary Of The Review:**

This paper considers a particular PU learning setting where the irreducibility assumption is not satisfied, and gives a new method to address this problem. The proposed method can be combined with any class-prior estimation method and the experimental results show positive effect of the proposed practical implementation. This paper is well-motivated and novel, and it will be better with the weaknesses addressed.

---

> ### Author Response · Authors · 2021-11-20
> **Response to Reviewer 6Jj8**
>
> _____________________________________________________________________________________________________________________
> + The parameter sensitivity analysis is needed in the experiment.
> + More related work about "irreducibility" can be introduced before Section 3. I don't know is there any solution for the irreducibility problem in mixture proportion estimation after reading the whole paper.
> _____________________________________________________________________________________________________________________
>
> **Q1. The parameter sensitivity analysis is needed in the experiment.**
>
> **A1.** If we understand the question correctly, the hyper-parameter $p$ sensitivity analysis is in Section 4.1. Figure 2 (b) shows that 1). our method has a similar estimation error compared to that of the baseline on the dataset satisfying the irreducibility assumption (the red curve) when $p \leq 0.15 $; 2). our method has a smaller estimation error compared to that of the baseline on the dataset dissatisfying the irreducibility assumption (the red curve)  when $p \leq 0.25 $.
>
> **Q2. More related work about "irreducibility" can be introduced before Section 3. I don't know is there any solution for the irreducibility problem in mixture proportion estimation after reading the whole paper.**
>
> **A2.** Thank you for pointing it out. We are sorry for the confusion. Only given unlabeled and positive data, without additional assumptions, existing methods contain bias if the assumption does not hold. However, if both positive and negative data are given, the linear Independence assumption can be used to replace the irreducibility assumption [1].
>
> Thank you very much for your other comments, we will further polish our paper for other minor issues.
>
> [1]. Yu, Xiyu, et al. "An efficient and provable approach for mixture proportion estimation using linear independence assumption." Proceedings of the IEEE Conference on Computer Vision and Pattern Recognition. 2018.

---

> > ### Comment · Reviewer_6Jj8 · 2021-11-24
> > **Response to rebuttal**
> >
> > Thanks for your responses.
> >
> > Figure 2 (b) does show some evidences about the selection of $p$ on synthetic data, while I think it would be more convincing if the hyper-
> > parameter sensitivity on (partial) real-world datasets can be conducted.
> >
> > Nevertheless, I still think this paper is above the acceptance threshold and I would like to keep my score.
> >
> >
> > Best Regards,
> >
> > Reviewer 6Jj8

---

### Official Review · Reviewer_JBHN · 2021-11-02

**Correctness:** 2
**Technical Novelty And Significance:** 2
**Empirical Novelty And Significance:** 2
**Recommendation:** 6
**Confidence:** 3

**Main Review:**

strengths:
1.Compared with the previous class-prior estimation methods, the authors consider the practical problems in the actual scene, that is, the distribution of the data set generally does not meet the ideal conditions. This is a relatively practical research topic.
2.In general, this article is well written, which is reflected in the clear organizational structure and concise and easy-to-understand presentation. The summary of related work is very informative.
3.The experimental design is generally reasonable, and the experiments with real -world data sets give more substantial results.
weaknesses:
1.Why is case-control PU learning more general than censoring PU learning? There is no clear explanation about this issue, please give a more detailed explanation in the part Introduction.
2.Regarding the main research scenarios in this article, that is, "assuming that the support of the positive class-conditional distribution is contained in the support of the negative class-conditional distribution", the article gives a theoretical definition. Can the author give a more intuitive explanation? Please give some more examples of this in actual problems. In what scenarios will such distributed data appear?
3.Why choose the most likely positive sample among the unlabeled samples? If the support of the positive class-conditional distribution is contained in the support of the negative class-conditional distribution, then intuitively these contained positive samples should look more like a negative sample.
4.In the ReCPE algorithm, the author proposes to find the most unlikely negative data among the unlabeled samples. The way to achieve this step is to train a binary classifier with the unlabeled sample and positive sample by treating unlabeled sample as a negative sample. When there are many positive samples in unlabeled samples, the "least unlikely negative sample" selected by the binary classifier obtained by this method is unreliable. Thereby the reliability of the next steps cannot be guaranteed.
5.The author mentioned that 10% was chosen as the value of the hyperparameter p in all experiments of the data set, but this approach seems to be contrary to Theory 1. This approach is too casual, I am worried that the new class-conditional distribution, which is obtained by transporting the probability mass of the set A from the negative class to the positive class, are really guaranteed to satisfy the irreduciblility assumption?
6.Why assume that the positive class-priorπis close to the new positive class-priorπ’? If the support of the positive class-conditional distribution is strongly contained in the support of the negative class-conditional distribution, The two class-priors should be quite different.
7.In general, although this research scenario is very meaningful, the ReCPE method proposed in this article is too strategic. It only proposes solutions intuitively, and there seems to be no strong guarantee in theory. The authors may be able to strengthen this article from this perspective.
Minor comments: The spelling of "real word" in Note 2 on page 7 is incorrect.



**Summary Of The Paper:**

This paper studies a relatively little-concerned problem, that is, the problem of class-prior estimation(CPE) in PU scenes. The authors concluded that the existing CPE methods are based on a critical assumption that the support of the positive data distribution cannot be contained in the support of the negative data distribution. However, in actual scenarios, this assumption may not be satisfied. It is also difficult to prove that a certain data set satisfies this assumption. Existing CPE methods will systematically overestimate the class prior when the data does not meet the critical assumption. To remove the assumption to make CPE always valid, a strategy called Regrouping CPE (ReCPE), which builds an auxiliary probability distribution, are proposed so that the support of the positive data distribution is never contained in the support of the negative data distribution. Theoretically, this method can give a more accurate estimate regardless of whether the dataset meets the assumption. The authors also proved the effectiveness of the method through a series of experiments.

**Summary Of The Review:**

This article focuses on practical issues, and its writing style is clear and easy to read. The structure of the article is reasonable, and the experiments are full and practical.
However, there are many parts that need to be improved in the method part, which are listed in Main review. In general, although this research scenario is very meaningful, the ReCPE method proposed in this article is too strategic. It only propose solutions intuitively, and there seems to be no strong guarantee in theory.

---

> ### Author Response · Authors · 2021-11-20
> **Response to Reviewer JBHN**
>
> ___________________________________________________________________________________________________________________
> + Why is case-control PU learning more general than censoring PU learning?
> + This paper assumes that the support of the positive class-conditional distribution is contained in the support of the negative class-conditional distribution.
> + Please give some more examples that the irreducibility assumption does not hold.
> + Why choose the most likely positive sample among the unlabeled samples?
> + When there are many positive samples in unlabeled samples, the "least unlikely negative sample" selected by the binary classifier obtained by this method is unreliable.
> + The author mentioned that 10% was chosen as the value of the hyperparameter p in all experiments of the data set, but this approach seems to be contrary to Theory 1. Is the algorithm really guaranteed to satisfy the irreducibility assumption?
> + Why assume that the positive class-prior $\pi$ is close to the new positive class-prior $\pi’$? If the support of the positive class-conditional distribution is strongly contained in the support of the negative class-conditional distribution, the two class-priors should be quite different.
> + The experimental settings such as network structures and hyper-parameters of Figure 2 are not clearly introduced neither in the supplementary material or main paper. Please clarify it.
> ___________________________________________________________________________________________________________________
> **Q1. Why is case-control PU learning more general than censoring PU learning?**
>
> **A1.** Thank you for the great question. Let $P_x$ be the marginal distribution of the data, and $P(X|Y=1)$ be the positive class conditional distribution.
>
> For case-control PU learning (two-sample case), the unlabeled sample $S_u$ and positive sample $S_p$ are directly drawn from two different distributions $P_x$ and $P(X|Y=1)$, respectively.
>
> For censoring PU learning (one-sample case), the positive sample is generated by selecting some positive examples from an unlabeled dataset with a constant probability, i.e., positive data are selected completely at random. The remaining unlabeled positive examples and all negative examples form an unlabeled sample.
>
> We show that the training data for censoring PU learning can be easily converted to the training data for case-control PU learning, but vice versa is hard. Therefore, case-control PU Learning is more general.
>
> To convert the training data for censoring PU learning to the training data for case-control PU learning, we could generate a new unlabeled sample following $P_x$ by simply adding the positive sample back to the unlabeled sample.
>
> To convert the training data for case-control PU learning to the training data for censoring PU learning, the union of the generated positive and unlabeled sample together must follow $P_x$. Then, the original positive sample has to be discarded. To generate a new positive sample, we have to select some positive examples completely at random among all the positive data in $S_u$. If so, we could already identify all the positive data in $S_u$ perfectly and could already reduce PU classification to PN classification. Therefore, case-control PU Learning is more general. It is also worth mentioning that, as a consequence, any method targeting case-control PU learning can be applied to case-control PU learning, but a method targeting case-control PU learning may not be applied to censoring PU learning without largely modifying its algorithm design. More details and proves can be found in [3].
>
> **Q2. This paper assumes that the support of the positive class-conditional distribution is contained in the support of the negative class-conditional distribution.**
>
> **A2.**  When the irreducibility assumption does not hold, the support of the positive class-conditional distribution is contained in the support of the negative class-conditional distribution. In our paper, we do not assume that the irreducibility assumption does not hold.
>
> The logic is that existing distribution-assumption-free CPE algorithms are designed based on the irreducible assumption. However, in practice, we can neither guarantee nor verify the irreducibility assumption. It implies that if directly employing existing CPE algorithms on PU datasets, those algorithms have a chance to give large estimation errors.
>
> To overcome the problem, we proposed a method that does not affect its base if the assumption already holds for the original probability distribution; otherwise, it reduces the positive bias of its base (see Theorem 2).

---

> > ### Author Response · Authors · 2021-11-20
> > **Response to Reviewer JBHN (Part 2)**
> >
> > **Q3. Please give some more examples that the irreducibility assumption does not hold.**
> >
> > **A3.** For example, estimating class priors from multiple sets of unlabeled data is a CPE problem that dissatisfied the irreducibility assumption. Specifically, we are given $m$ $(m \geq 2)$ sets of unlabeled samples drawn from different marginal densities. Each unlabeled sample $S_{u_i}$ follows density $p_i = \pi_i p_p+ (1-\pi_i)p_n$, where $ i \in \\{2, m\\}$. Only given those sets of samples, it is theoretically impossible to learn class priors, and existing methods assume all necessary class priors are given [1, 2]. Similar in learning with label noise [4], when the label noise does not dependent on instances. Estimating noisy class-posterior is also a CPE problem that dissatisfied the irreducibility assumption.
> >
> > **Q4. Why choose the most likely positive sample among the unlabeled samples?**
> >
> > **A4.**  In Theorem 4, we have proved that the most likely positive sample should be chosen to reduce the bias of existing methods. Intuitively, by copying examples from unlabeled data to positive data, we have created an auxiliary positive distribution. Choosing the most likely positive sample among the unlabeled samples encourages the difference between the original and auxiliary positive distributions to be small, but at the same time, the irreducibility assumption is satisfied. Then the estimated class prior is close to the original class prior.
> >
> > **Q5. When there are many positive samples in unlabeled samples, the "least unlikely negative sample" selected by the binary classifier obtained by this method is unreliable.**
> >
> > **A5.**  When the sample size is small, the generalization errors of all existing methods are large, and all methods produce large estimation errors. The unreliable comes from the large difference between empirical distribution and the underlying distribution of negative data but not our method.
> >
> > **Q6. The author mentioned that 10% was chosen as the value of the hyperparameter $p$ in all experiments of the data set, but this approach seems to be contrary to Theory 1. Is the algorithm really guaranteed to satisfy the irreducibility assumption?**
> >
> > **A6.** We have also found this interesting phenomenon. Theoretically, we should encourage the set used for regrouping to be small. However, empirically, existing estimators are insensitive to tiny modifications. We think that they are designed to be robust in such a way, in order to be good estimators. Then, $p$ cannot be so small in practice.
> >
> > Here we explain that the reason our method satisfies the irreducibility assumption. Theoretically, we have transferred all the probability mass of a set A to the positive class. It implies that the probability of set a in new negative class is $0$, but it is not $0$ in positive class. Therefore, the assumption is guaranteed to be satisfied.
> >
> > Empirically, by copying a set of examples (set A) from unlabeled data to positive data, we have created a new positive sample. By comparing unlabeled and new positive samples, we could find that each example in set A appears in both unlabeled and positive samples with the same frequency. it implies that all these examples in set A in unlabeled data are drawn from new positive distribution but not new (latent) negative distribution. Then support of positive distribution is not in the support of the negative distribution, which also satisfies the irreducibility assumption.
> >
> > **Q7. Why assume that the positive class-prior $\pi$ is close to the new positive class-prior $\pi’$? If the support of the positive class-conditional distribution is strongly contained in the support of the negative class-conditional distribution, the two class-priors should be quite different.**
> >
> > **A8.** For all experiments, we employ a neural network with $2$ hidden layers. Each hidden layer contains $50$ hidden units. The batch normalization is also employed. The stochastic gradient descent optimizer is used with the batch size $50$. The network is trained for 350 epochs with a learning rate $0.01$ and momentum $0$. The weight decay is set to $1e-5$. The model with the best validation accuracy is used to estimate the positive class-posterior probability $P(Y=1|X=x)$ (See Section 4). We sample the validation set with 20\% of the training data size. The hyper-parameters vary from 0 to 0.25 with a step size 0.05.

---

> > > ### Author Response · Authors · 2021-11-20
> > > **Response to Reviewer JBHN (Reference)**
> > >
> > > [1]. Lu, Nan, et al. "Mitigating overfitting in supervised classification from two unlabeled datasets: A consistent risk correction approach." International Conference on Artificial Intelligence and Statistics. PMLR, 2020.
> > >
> > > [2]. Lu, Nan, et al. "Binary Classification from Multiple Unlabeled Datasets via Surrogate Set Classification." International Conference on Machine Learning. PMLR, 2021.
> > >
> > > [3]. Menon, Aditya, et al. "Learning from corrupted binary labels via class-probability estimation." International Conference on Machine Learning. PMLR, 2015.
> > >
> > > [4]. Patrini, Giorgio, et al. "Making deep neural networks robust to label noise: A loss correction approach." Proceedings of the IEEE conference on computer vision and pattern recognition. 2017.

---

### Official Review · Reviewer_EaHM · 2021-11-04

**Correctness:** 4
**Technical Novelty And Significance:** 3
**Empirical Novelty And Significance:** Not applicable
**Recommendation:** 8
**Confidence:** 4

**Main Review:**

**Strengths**
- The paper is well-written and easy to follow. The mathematical contents are not hard to parse and examples are given.
- This work is well-motivated and novel. The paper addresses the overestimation problem of $\pi$ in PU learning by proposing a regrouping technique.
- Both the theoretical and experimental proofs are given. They help support the authors' claims.
- Although the gain is small in absolute value, the large number of repeat experiments makes it statistically significant (and statistical tests are included).

**Weaknesses and Clarifications**
- The authors exclusively compare the estimation of $\pi$. While I understand that the method is aimed at estimating the prior, I think that it would be beneficial if the authors can provide the PU learning results with different priors (fixed, CPE, ReCPE, etc. ). A simple nnPU would suffice.
- Many "irreducibility" has a typo ("irreduciblility").


**Summary Of The Paper:**

This paper studies the prior $\pi$ in PU learning. More specifically, it studies the estimation method of the prior $\pi$ in a more general case, i.e., without the irreducibility assumption. When unsatisfied, such an assumption will often lead to the overestimation of $\pi$. However, current class-prior estimation methods are usually based on the irreducibility assumption. The authors propose a new CPE problem based on a new auxiliary distribution that always satisfies the irreducibility requirement. The technique is called regrouping, which transforms the true distributions to  Both theoretical and experimental results are included in the paper, showing the validness of the proposed ReCPE method.

**Summary Of The Review:**

The authors give an insightful demonstration of limitations of existing CPE methods, and also give feasible solutions to the limitations. Theoretical proofs are given, and experiment results show significant superiority of the proposed method compared to existing CPE methods. So I believe this work would be beneficial to the community and therefore give a score of 8.

---

> ### Author Response · Authors · 2021-11-20
> **Response to Reviewer EaHM**
>
> _____________________________________________________________________________________________________________________
> + I think that it would be beneficial if the authors can provide the PU learning results with different priors (fixed, CPE, ReCPE, etc. ).
> _____________________________________________________________________________________________________________________
>
> **Q1. I think that it would be beneficial if the authors can provide the PU learning results with different priors.**
>
> **A1.**  Thank you for the great suggestion. We will provide PU learning classification results by using different PU methods in the supplementary martial.

---

> > ### Comment · Reviewer_EaHM · 2021-11-29
> > **Final score**
> >
> > Thank you for the authors' feedback. I decide to keep my score after reading other reviewers' comments and the authors' response.

---

### Official Review · Reviewer_8SWo · 2021-11-07

**Correctness:** 4
**Technical Novelty And Significance:** 2
**Empirical Novelty And Significance:** 2
**Recommendation:** 5
**Confidence:** 4

**Main Review:**

+++Strength:
1. PU learning is apparently an important machine learning problem, which has the potential to be used in many real-world applications.
2. The analyses on the prior studies appear reasonable.
3. Experiments have been conducted to support the proposed method and the results appear good.


---Weakness:
1. While the proposed probability distribution transformation makes sense to avoid relationship between the prior of positive data and that of negative data, the proposed method appear pretty simple. This might not be a weakness, but methodology is not a strength of this paper.
2. A big gripe of this paper is its writing. There exist too many grammatical/syntactical mistakes. Here I list few in the below.

Page 2: "To let these distributional-assumption-free methods can be used to identify class-prior $\pi$, $\kappa^*$ must ...". This sentence can be definitely written in a better way.

Page 2: "Because the irreduciblility assumption is impossible to check without making any assumption on $P_n$. Thereby, in ...". This is an incomplete sentence, which is a mistake.

Page 3: "... by creating a new auxiliary distribution $P_{p'}$ always guaranteeing the irreduciblility assumption..." Here using an attributive clause is much better than using a present participle.

Page 4: "... i.e., there exists a set can achieve the minimum 0, ..." This sentence is simply wrong.

3. As we all know, deep learning has become the main stream of machine learning in the past decade, which is particularly true in ICLR. Many research on PU learning using deep neural networks have been proposed and demonstrated good performance. As a result, without comparing to such methods, it is harder for reader to appreciate the effectiveness of the paper.

**Summary Of The Paper:**

This paper studies positive and unlabeled (PU) learning, which apparently is an important problem in machine learning. In many existing researches, the PU classifiers assumes that the irreducibility assumption holds on input data, which, though, may not always be true in many real-world applications. As a result, the learn classifier might not work well due to estimation bias on data prior. To address this problem, this paper proposes to construct a transformed the probability distributions of input data by a re-grouping operation, such as the positive prior cannot be a component (or called as a support) of the negative prior. Good experiment results have been obtained to support the proposed method.

**Summary Of The Review:**

This paper studies positive and unlabeled (PU) learning by constructing a transformed the probability distributions of input data by a re-grouping operation, such as the irreduciblility assumption holds on the data to the subsequent classifier. Overall, this paper is clearly written. But because of the following concerns, also mentioned above, I suggest the authors to further improve this paper before publication:
1. Please proofread this manuscript carefully to improve the writing.
2. More comprehensive experiments to compare deep neural network based PU learning should be added to convince the readers.

---

> ### Author Response · Authors · 2021-11-20
> **Response to Reviewer 8SWo**
>
> ____________________________________________________________________________________________________________________
> + The proposed method appears pretty simple. This might not be a weakness, but methodology is not a strength of this paper.
> + There exist too many grammatical/syntactical mistakes.
> + Without comparing to the methods using deep neural networks, it is harder for reader to appreciate the effec-tiveness of the paper.
> ____________________________________________________________________________________________________________________
>
> **Q1. The proposed method appears pretty simple. This might not be a weakness, but methodology is not a strength of this paper.**
>
> **A1.** This is the first paper that proposed a method to solve CPE without the irreducibility assumption. The bias of our method has also been theoretically justified. We also try to make our algorithm as simple and general as possible such that it can be easily integrated into existing CPE methods. Here we would like to emphasize the problem and contributions.
>
> Existing distribution-assumption-free CPE methods are designed based on the irreducibility assumption (including SOTA algorithms). However, in practice, we cannot *guarantee* and *verify* the assumption based on the data. As a result, existing methods would give much worse results if the irreducibility assumption does not hold.
>
> To solve the issue, we change the original CPE to a new CPE with irreducibility. Theoretically, we have proved that our method does not affect its base if the assumption already holds for the original probability distribution; otherwise, it reduces the positive bias of its base (see Theorem 2). This is the major contribution of our paper.
>
> Algorithm 1 can approximately obtain the transformation, we have proved in Proposition 2. Specifically, Proposition 2 shows the difference between the obtained approximate transformation and the true transformation is upper bounded by $O(\epsilon)$, and we can control $\epsilon$ to be small in practice.
>
> Empirically, synthetic datasets (where their ground truth are known) are used to validate the correctness of our claims. Figure 2a clearly shows that: 1). the error $\epsilon$ can be controlled by a hyperparameter $p$; 2). On the dataset dissatisfying the irreducibility assumption, $\pi'$ could be smaller than the maximum proportion $\kappa^*$; on the dataset satisfying the irreducibility assumption (irreducible data), $\pi'$ should be close to $\pi$.
>
> **Q2. There exist too many grammatical/syntactical mistakes.**
>
> **A2.** Thank you for your constructive comments. We have carefully revised and updated the manuscript.
>
> **Q3. Without comparing to the methods using deep neural networks, it is harder for reader to appreciate the effectiveness of the paper?**
>
> **A3.** Thank you very much for pointing this out. We are sorry for the confusion. Both baseline methods alphaMax [1] and DEDPUL [2] use deep neural networks. We will add more introductions for each baseline method in the supplementary martial.
>
> [1]. Jain, Shantanu, et al. "Nonparametric semi-supervised learning of class proportions." arXiv preprint arXiv:1601.01944 (2016).
>
> [2]. Ivanov, Dmitry. "DEDPUL: Difference-of-Estimated-Densities-based Positive-Unlabeled Learning." 2020 19th IEEE International Conference on Machine Learning and Applications (ICMLA). IEEE, 2020.

---

> ### Author Response · Authors · 2021-11-24
> **Discussion**
>
> Dear Reviewer 8SWo,
>
> We appreciate your comments and time! We have revised the paper following your suggestions. Would you mind checking it and confirming if you have further questions?
>
> Best Regards,
> Paper435 Authors

---

### Decision · Program_Chairs · 2022-01-20

**Decision:**

Accept (Poster)

**Comment:**

This paper received a majority vote for acceptance from reviewers and me. I have read all the materials of this paper including manuscript, appendix, comments and response. Based on collected information from all reviewers and my personal judgement, I can make the recommendation on this paper, *acceptance*. Here are the comments that I summarized, which include my opinion and evidence.

**Research Motivation and Problem**

This paper is well motivated by the agnostic of CPE assumption that the support of the positive data distribution cannot be contained in the support of the negative data distribution. To tackle this problem, the authors built an auxiliary probability distribution such that the support of the positive data distribution is never contained in the support of the negative data distribution.

**Technical Contribution**

The technical part is simple and clear. The regrouping idea is also easy to implement. The theoretical justification is a good complement of the proposed ReCPE algorithm.

**ReCPE does not affect its base if the assumption already holds**

The authors employed the synthetic datasets to verify this point. This is a plus.

**Experimental Results**

The authors demonstrated their ReCPE algorithm can be used as a booster on seven base PU classifiers.

**Presentation**

The presentation has been much improved with the guidance of one reviewer. But I found two extra minor ones. (1) "PU Learning" -> "PU learning" at the beginning of the second paragraph on Page 1. (2) Two typos in "9 real word datasets." on Page 7 -> "9 real-world dataset.", where the footnote should be placed after the period.

**Layout**

(1) Too many lines in Table 1. It is suggested to remove the horizontal lines among the same dataset, (2) Appendix should go after the main manuscript, rather than a separate file.

No objection from reviewers was raised to again this recommendation.